# A Recurrent Adaptive Network: Balanced Learning for Road Crack Segmentation with High-Resolution Images

Yi Zhang [1,2], Junfu Fan [1,3,*], Mengzhen Zhang [1], Zongwen Shi [1], Rufei Liu [4] and Bing Guo [1]

1   School of Civil and Architectural Engineering, Shandong University of Technology, Zibo 255000, China; zhangyi_csu@csu.edu.cn (Y.Z.); zhangmz@lreis.ac.cn (M.Z.); 21407010766@stumail.sdut.edu.cn (Z.S.); guobing@sdut.edu.cn (B.G.)
2   School of Geosciences and Info-Physics, Central South University, Changsha 410083, China
3   State Key Laboratory of Resources and Environmental Information System, Chinese Academy of Sciences, Beijing 100101, China
4   College of Geodesy and Geomatics, Shandong University of Science and Technology, Qingdao 266590, China; liurufei@sdust.edu.cn
*   Correspondence: fanjf@sdut.edu.cn

**Abstract:** Road crack segmentation based on high-resolution images is an important task in road service maintenance. The undamaged road surface area is much larger than the damaged area on a highway. This imbalanced situation yields poor road crack segmentation performance for convolutional neural networks. In this paper, we first evaluate the mainstream convolutional neural network structure in the road crack segmentation task. Second, inspired by the second law of thermodynamics, an improved method called a recurrent adaptive network for a pixelwise road crack segmentation task is proposed to solve the extreme imbalance between positive and negative samples. We achieved a flow between precision and recall, similar to the conduction of temperature repetition. During the training process, the recurrent adaptive network (1) dynamically evaluates the degree of imbalance, (2) determines the positive and negative sampling rates, and (3) adjusts the loss weights of positive and negative features. By following these steps, we established a channel between precision and recall and kept them balanced as they flow to each other. A dataset of high-resolution road crack images with annotations (named HRRC) was built from a real road inspection scene. The images in HRRC were collected on a mobile vehicle measurement platform by high-resolution industrial cameras and were carefully labeled at the pixel level. Therefore, this dataset has sufficient data complexity to objectively evaluate the real performance of convolutional neural networks in highway patrol scenes. Our main contribution is a new method of solving the data imbalance problem, and the method of guiding model training by analyzing precision and recall is experimentally demonstrated to be effective. The recurrent adaptive network achieves state-of-the-art performance on this dataset.

**Keywords:** road crack segmentation; convolutional neural networks; balanced deep learning; adaptive loss

## 1. Introduction

Road crack segmentation has important application value in road maintenance, especially for highways. It is challenging for road management departments to quickly determine the large-scale technical conditions of asphalt road pavement because of the rapid increase in road network mileage [1]. Scholars proposed various traditional computer vision-based road crack segmentation methods before convolutional neural networks became mainstream. Traditional digital image processing (DIP) methods, such as the Canny edge detector [2], wavelet transform [3], and crack index [4], have been carried out for crack segmentation. However, these algorithms focus on a small group of pixels and lack an understanding of the image content. Notably, DIP-based methods are very sensitive to noise [5], such as shadows and stains, and are unable to meet the needs of large-scale road

maintenance. In the past five years, methods for extracting road cracks via convolutional neural networks have gradually replaced traditional methods and have become mainstream approaches in scientific research. Deep learning methods can better preserve the geometry of detected crack patterns, and their predictions (in terms of the pixels belonging to a crack) have ultimately become more accurate than those of the threshold method [6]. Therefore, the DIP method is not discussed in the following sections.

With the development of deep learning technology, convolutional neural networks such as ResNet [7] and DeepLabV3 [8] have achieved great success in computer vision and have been naturally introduced to the road crack segmentation task. Due to their powerful ability to learn high-level feature representations, road crack segmentation approaches using supervised convolutional neural networks have achieved good results. However, most studies have only applied advanced computer vision methods to the field of road damage segmentation and have not deeply considered the differences between road cracks and natural images. Compared with the semantic segmentation task involving natural images, the road crack segmentation task has unique characteristics:

- Small targets: According to the latest "Highway Technical Condition Evaluation Standard JTG 5210-2018", cracks with average widths greater than 1 mm should be extracted. Many road crack instances are only two or three pixels wide, making them much smaller than natural images in datasets such as PASCAL Visual Object Class (VOC), and Microsoft Common Objects in Context (MS-COCO), even in very high-resolution images.
- Imbalance at the sample level: The positive and negative data are imbalanced at the sample level. For example, in the DeepCrack dataset [9], the ratio of positive and negative samples is approximately 1:33, and in the HRRC dataset, the ratio greatly increases to 1:705 at the pixel level. The randomness and sparsity of the positive class are even more apparent in the engineered HRRC dataset.
- Imbalance at the feature level: Driven from samples, imbalance also occurs at the feature level. Deep high-level features in backbones have more semantic meanings, while shallow low-level features are more descriptive in terms of content [10]. The gradient generated by a small number of positive features, especially at a deep level, may be submerged by the gradient generated by a large number of negative features, and the utilized model will have difficulty achieving enough positive characteristics.

Due to the problems above, extracting road cracks from real road inspection images is still a challenging task. A neural network method based on supervised learning cannot achieve good results in real road inspection scenarios, as shown in Table 1.

In the case of supervised learning, the small target and imbalance problems greatly reduce the ability of convolutional neural networks to perform road crack segmentation. We are in urgent need of a simple, universal, and effective method to solve such problems and meet the needs of engineering applications. Two questions need to be rethought: what network structure is best suited for small target recognition? How can the extreme imbalance problem be solved with a stable, universal, and adaptive solution?

To solve the first question, we tested the current dominant computer vision-based convolutional neural network architecture in real road inspection scenarios to determine a powerful baseline network. To solve the second problem, we follow a simple line of thought: a poor positive sample recognition ability will result in high precision and low recall. Can we obtain better results if we maintain a relative balance between precision and recall? The second law of thermodynamics states that heat can be spontaneously transferred from a hotter body to a cooler body but cannot be spontaneously transferred from a cooler body to a hotter body. Inspired by this, we put forward a main point of view: is it possible to establish a channel between precision and recall in which indicators flow dynamically like temperature? Therefore, we propose a precision–recall–flow (PRF) index to adaptively control the strength and direction of the flow and take measures at the sample level and feature level to drive the flow recurrently.

In this paper, our main contributions are as follows. (1) We review and analyze several kinds of convolutional neural networks. The experiments were performed to prove that the HRNet [11] architecture with rich high-resolution representation has advantages in road crack extraction tasks. By replacing batch normalization with group normalization (HRNet) and group normalization (HRNet-GN), the network performance was improved. (2) We propose a recurrent adaptive network for equilibrium learning. The network treats precision and recall as temperatures in different regions of the model and forces energy to flow from high- to low-temperature regions by automatically adjusting sampling and loss weights. The proposed method balances precision and recall and improves the ability of the model to handle unbalanced data. (3) A new high-resolution road crack database named HRRC is established. Its road images were continuously captured by high-resolution cameras during the driving process. The road crack areas in these images were carefully labeled by LabelMe [12] and were sliced into 24,704 images, each with dimensions of $512 \times 512$ pixels. The dataset has abundant image noise, such as shadows, stains, filling joints, and cars, and was used to verify the capability of the proposed method. This paper presents a new way of addressing data imbalance in deep learning tasks by analyzing precision versus recall and represents the first study that attempts to measure data imbalance in terms of the relationship between precision and recall rather than the data ratio.

The remainder of this paper is organized as follows. In Section 2, we introduce some existing convolutional neural network structures for segmentation and methods for solving the imbalance problem. Section 3 describes the proposed method. The experimental details and results are reported in Sections 4 and 5, our findings and future works are discussed in Section 6, and this paper is concluded in Section 7.

## 2. Related Work

Road crack segmentation methods can be divided into two categories: traditional digital image processing and machine learning-based approaches. In the past five years, machine learning-based crack segmentation methods have repeatedly used advanced state-of-the-art techniques [9,13] and have replaced traditional methods. In this section, we review works involving convolutional neural networks for solving network architecture and imbalance problems. The road crack segmentation dataset used in the mainstream study is presented.

### 2.1. Network Architectures

The fully convolutional network (FCN) [14] architecture transforms fully connected layers into convolution layers, enabling a classification network to output a heatmap. A convolutional layer is used after the pooling layer to convert the fully connected layers in VGG16 [15] and up-sample the predictions back to pixels in a single step. The FCN architecture builds "fully convolutional" networks that take inputs of an arbitrary size and produce correspondingly sized outputs with efficient inference and learning processes.

With an FCN architecture, Chun [16] proposed a fully convolutional neural network-based road surface damage detection approach with semisupervised learning. The model is updated by using pseudolabeled images derived from semisupervised learning methods to improve the performance of road surface damage detection techniques.

The U-Net [17] architecture achieves very good performance in very different biomedical segmentation applications and has become one of the most commonly used network structures. The architecture consists of a contracting path to capture context and a symmetric expanding path that enables precise localization. Using skip connections between the two parts, U-Net can mix shallow, low-level feature maps obtained from the encoder and deep, semantic features derived from the decoder. Benefiting from skip connections, the network can better retain low-dimensional features with high resolution to achieve good results in medical segmentation tasks with rich details, such as boundaries and seams.

With the U-Net architecture, Rodrigo [18] proposed a novel synthetic dataset and a weakly supervised learning method to overcome the inaccurate annotation problem

and improved the results by up to 12%. Hong [19] proposed an improved identification technique based on the U-Net architecture that was enhanced with a convolutional block attention module, an improved encoder, and the strategy of fusing long- and short-skip connections. This method could effectively predict highway cracks in unmanned aerial vehicle (UAV) images.

UNet++ [20] alleviates the unknown network depth problem with an efficient ensemble of U-Nets with varying depths, which partially share an encoder and simultaneously perform co-learning using deep supervision. Furthermore, the skip connections are redesigned to aggregate features of varying semantic scales at the decoder. These network structures effectively preserve shallow features and are widely used in medical image segmentation.

With the UNet++ architecture, Yang [21] proposed a method composed of two stages, crack recognition and crack semantic segmentation, making it easy to meet the needs of efficient and reliable detection for large-scale collected images. The fine-tuned VGG16 model in the first stage can accurately identify crack images and avoid the computing costs incurred by the further processing of non-crack images, and the UNet++ model in the second stage provides pixel-level semantic segmentation for crack images.

The pyramid scene parsing network (PSPNet) [22] was proposed to embed difficult scenery context features in an FCN-based pixel prediction framework. The pyramid pooling module fuses features under four different pyramid scales, aggregates multilevel context to global context information, and provides global-scene-level information to the developed model. This architecture exploits the capability of global context information derived from different-region-based context aggregation through a pyramid pooling module and a suitable strategy to utilize global scene category clues.

A feature pyramid network (FPN) [23] naturally leverages the pyramidal shape of a ConvNet's feature hierarchy while creating a feature pyramid that has strong semantics at all scales. An FPN relies on an architecture that combines low-resolution, semantically strong features with high-resolution, semantically weak features via a top-down pathway and lateral connections, thereby developing a top-down architecture to build high-level semantic feature maps at all scales.

With a similar network structure, DeepCrack [9] resizes, concatenates, and fuses these multiscale feature maps at all scales, and a guide filter [24] is used to perform denoising at the feature level before prediction. APLCNet [25] combines a channel attention mechanism and a spatial attention mechanism during FPN feature fusion, highlights the attribute information and location information of cracks, and achieves good results on road crack instance segmentation tasks.

DeepLabV3+ [26] improves the encoder–decoder structure of other approaches. The encoder module encodes multiscale contextual information by applying atrous convolution on multiple scales, while the simple and efficient decoder module refines the segmentation results along the target boundary. The Xception model is further explored for the segmentation task. Deep separable convolution is applied to the ASPP and decoder modules to produce faster and stronger encoder–decoder networks.

A high-resolution network (HRNet) [11] maintains high-resolution representations by connecting high-to-low resolution convolutions in parallel and strengthens high-resolution representations by repeatedly performing multiscale fusions across parallel convolutions. HRNet has achieved great success in key point detection, attitude estimation, and multiperson attitude estimation tasks and exhibits enormous potential for scientific research and applications.

Based on HRNet, Chen [27] proposed an enhanced version called HRNete by removing the down-sampling operation in the initial stage, reducing the number of high-resolution representation layers, using dilated convolution, and introducing hierarchical feature integration. Using HRNet as the backbone, Bai [28] proposed the robust mask R-CNN, an end-to-end deep neural network for crack detection and segmentation on structures or their components that may be damaged during extreme events, such as earthquakes.

### 2.2. Imbalance Problem

Libra R-CNN [29] utilizes a simple but effective framework for balanced learning. It integrates IOU-balanced sampling by splitting the sampling interval into K bins according to their IOU measures and uniformly selects samples from these bins. It obtains a balanced feature pyramid by resizing the multilevel features in an FPN, averaging them to obtain an integrate feature, making refinements with convolutions, and resizing the features to their original sizes. A balanced L1 loss is set by separating inliers from outliers and clipping the large gradients produced by outliers with a maximum value of 1.0.

A two-stage convolutional neural network [13] was proposed for road crack detection and segmentation in images at the pixel level. The first stage serves to remove noise or artifacts and isolate the potential cracks in a small area via a classification network that is composed of five-layer convolutional neural networks and two fully connected (FC) layers. The second stage is a U-Net-structured segmentation network that can learn the context of cracks in the detected area. In this two-step framework, the first network filters out pure negative sample images, and images containing positive samples are allowed to proceed to the second stage; this approach can reduce the proportion of negative samples.

Focal loss [30] addresses the class imbalance problem by reshaping the standard cross-entropy loss such that it down-weights the losses assigned to well-classified examples. Focal loss focuses training on a sparse set of hard examples and prevents the vast number of easy negatives from overwhelming the detector during training. Two preset hyperparameters are employed; one is used to reduce the loss contributions provided by easy examples and extend their range, and the other addresses class imbalance.

Dice loss [31] considers that predictions are strongly biased toward the background if the area of interest occupies only a very small region of an image, and a loss function based on the Dice coefficient (related to precision and recall) can ameliorate this situation. The F1 score considers both the precision and recall of classification models and can comprehensively describe the performance of a model. The Dice loss is defined as 1 minus the F1 score, which involves maximizing the value of the F1 score as the optimization condition.

RAO-UNet [32] proposed a foreground perception optimization method. This method calculates the ratio of the sum of the probabilities of all pixels predicted as the background to the sum of the probabilities of all pixels predicted as the foreground in the predicted probability map. This ratio is used as an adaptive parameter for weighted binary cross entropy (BCE) loss in each batch to improve the performance damage caused by data imbalance.

Generative adversarial networks (GANs) [33] can extend the data by generating virtual data to solve the data imbalance problem. This approach has been used to deal with the problem of unbalanced landslide remote sensing data [34]. In Ref. [35], crack detection was enhanced by a generative adversarial network. CrackGAN [36] uses the generator as a segmentation network and adds a new constraint, generative adversarial loss, to regularize the objective function, which makes the network always generate a crack-GT detection result. These studies demonstrate the effectiveness of GANs in combating sample imbalance.

The technique of randomly under-sampling the majority class (RUMC) [37] involves randomly selecting examples from the majority class and removing them for the training dataset. The majority-class instances are discarded at random until a balanced class distribution in the training set is reached.

### 2.3. Dataset Situation of Road Crake Segmentation

At present, existing road crack segmentation datasets, such as DeepCrack [9], Crack-Forest [38], and 2StageCrack [13], are used by researchers to conduct experiments. The DeepCrack dataset contains 537 images with dimensions of $544 \times 384$ pixels. The Crack-Forest dataset contains 118 labeled images of $544 \times 384$ pixels. The 2StageCrack dataset, the newest open-source dataset, contains 1276 images for training and 354 for testing, each with dimensions of $96 \times 96$ pixels.

## 3. The Proposed Method

The main idea of our method is inspired by the second law of thermodynamics: heat can be spontaneously transferred from a hotter body to a cooler body. We hope to establish a similar mechanism during the convolutional neural network training process to regard precision and recall as body temperature and make them naturally flow from high to low. Therefore, we need to solve three problems: how to determine the direction of the flow, how to determine the strength of the flow, and how to make the precision and recall flow spontaneously.

To overcome these problems, our proposed method is as follows. (1) We define an adaptive parameter called PRF, which is associated with precision and recall, to evaluate the gap between precision and recall at a defined interval. (2) The flow direction is determined according to the positive and negative values of the PRF, and the flow intensity is determined according to the absolute value of the PRF. (3) To bridge precision and recall at the sample level and feature level, the sampling method and loss weight are repeatedly adjusted during the whole training process. Finally, spontaneous flow and a dynamic balance between precision and recall are achieved to narrow the gap between them and obtain a better trained model. The overall design of the recurrent adaptive network framework is shown in Figure 1.

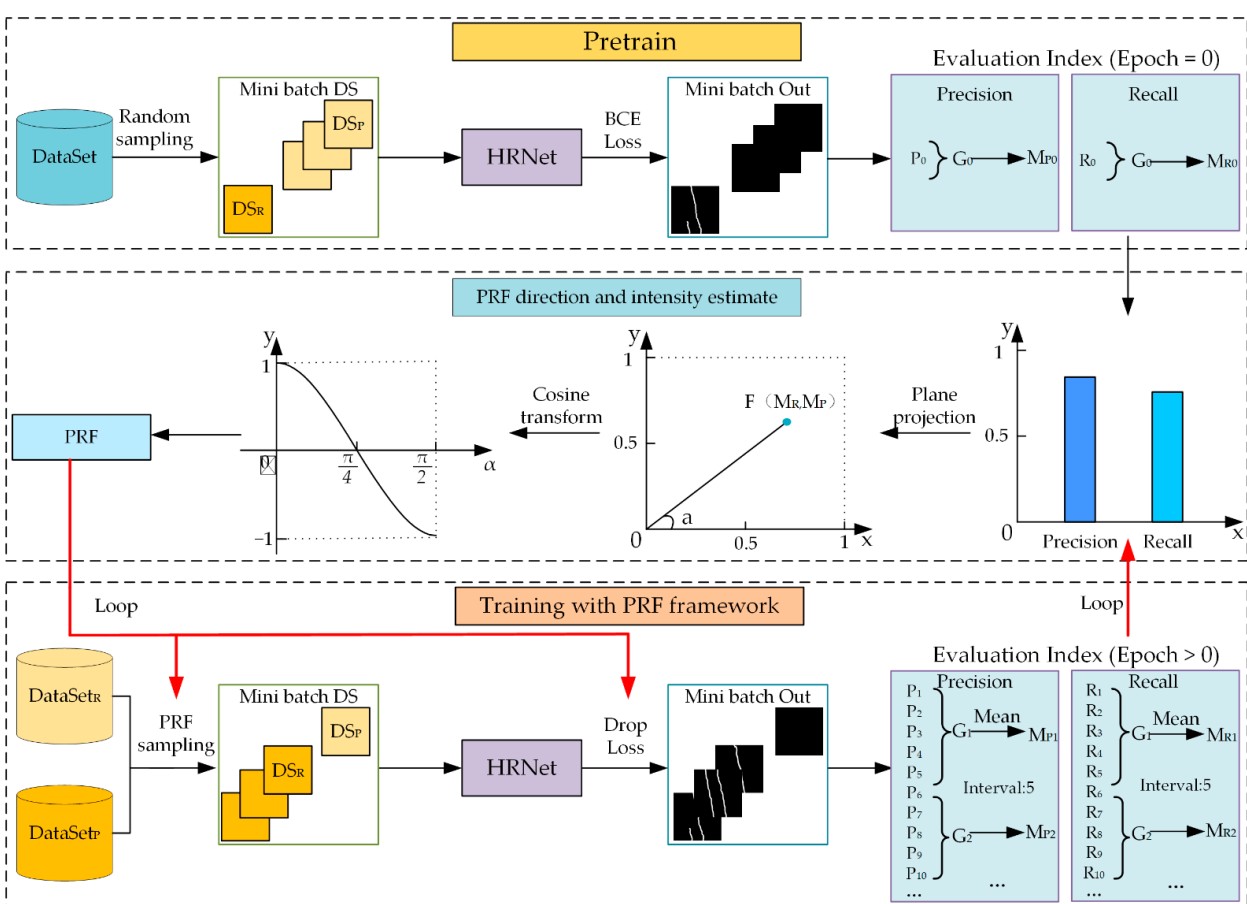

**Figure 1.** The overall design of the recurrent adaptive network framework.

### 3.1. Precision–Recall Flow Definition

During the training process, the average values of precision and recall during each epoch are calculated and added to the list L ($R_i$, $P_i$). To reduce the deviation caused by occasional shocks during the training procedures, we organize the values in L into groups at a certain interval and calculate the average precision and recall values in each group to obtain $L_g$ ($M_{Ri}$, $M_{Pi}$). Restricted to supervised training, the offset degree can be displayed

intuitively in $L_g$ with an imbalanced dataset—there will be a gap between recall and precision. As the degree of data imbalance increases, this gap will also become larger. A rectangular coordinate system is plotted for each pair of values in $L_g$. Point F ($M_R$, $M_P$) is obtained with its corresponding recall and mean precision as the coordinates, and the azimuth of line OF is calculated as:

$$\alpha = \arctan(\frac{M_P}{M_R}), M_P \in [0,1], M_R \in [0,1] \tag{1}$$

where $M_P$ is the average value of precision at interval, $M_R$ is the average value of recall at interval, and $\alpha$ is the angle between OF and *X*-axis in Figure 1.

The *PRF* is defined as the evaluation index:

$$PRF = \cos(2\alpha), \alpha \in \left[0, \frac{\pi}{2}\right], PRF \in [-1,1] \tag{2}$$

### 3.2. Precision–Recall Flow Direction and Intensity Estimate

In different datasets, the proportions of positive and negative samples are different. Negative samples, which form the larger imbalanced class during the supervised learning of road crack segmentation, overwhelm the BCE loss, Dice loss, and Focal loss. To avoid the lopsidedness of precision and recall, we need to evaluate the model's bias toward precision or recall, measure the degree of such an imbalance, and improve it.

The PRF parameters are obtained by cosine transformation, which can effectively meet the requirements of determining the flow direction and intensity; the direction of the flow is indicated by its positive or negative sign and the intensity of the flow is measured by its absolute value. When the positive sample recognition ability is strong, the precision is lower than the recall. In contrast, a precision higher than the recall means that the model is more advantageous in identifying negative samples. Therefore, the range of the PRF can clearly indicate the flow direction of precision and recall and can be expressed by the following equation:

$$\begin{array}{cc} M_P > M_R & PRF \in (0,1] \\ M_P = M_R & PRF = 0 \\ M_P < M_R & PRF \in [-1,0) \end{array} \tag{3}$$

where $M_P$ is the average value of precision at interval and $M_R$ is the average value of recall at interval.

Furthermore, we need to estimate the strength of the flow; the gap between the precision and recall should be proportional to the flow strength. According to the calculation process, when the distance between precision and recall approaches 0, the absolute value of the PRF approaches 0, and when the distance approaches 1, the absolute value of the PRF approaches 1. Therefore, the PRF can measure the degree of deviation: the larger the absolute value of the PRF is, the more serious the deviation is and a greater flow intensity is needed. The PRF can effectively solve the problem of intensity estimation.

### 3.3. Precision–Recall Flow Sampling Method

When training a model, a random sampling scheme usually leads to the selected samples being dominated by negative samples in the road crack segmentation task, as shown in Table 1. To avoid this situation, we propose the PRF sampling method. The dataset is divided into two subsets according to the proportion of the positive sample pixels of each image; the images with positive sample pixels accounting for more than 0.5% of the total are put into dataset R, and the rest are placed into dataset P. We sample from the two subsets at different probabilities (controlled by the parameter, S). The initial value of S is 0.1; iterating with the PRF produces updates, as shown in the following equation:

$$S = \begin{cases} 0.01 & S < 0.01 \\ S(1 + PRF) & 0.2 < S < 0.01 \quad S \in [0.01, 0.2] \\ 0.2 & S > 0.2 \end{cases} \tag{4}$$

where *S* is the probability of sampling from dataset P.

The sampling strategy is as follows. Before each sampling step, take a random number R∈ [0,1]. When R is less than *S*, the sample is from dataset P; otherwise, the sample is from dataset R, as shown in Figure 2. In this way, dominant classes are suppressed during sampling, and smaller classes are more likely to enter the network. *S* is cyclically and continuously modified to ensure that balance is maintained at each stage. As a result, the precision and recall will flow from higher to lower at the data sampling level.

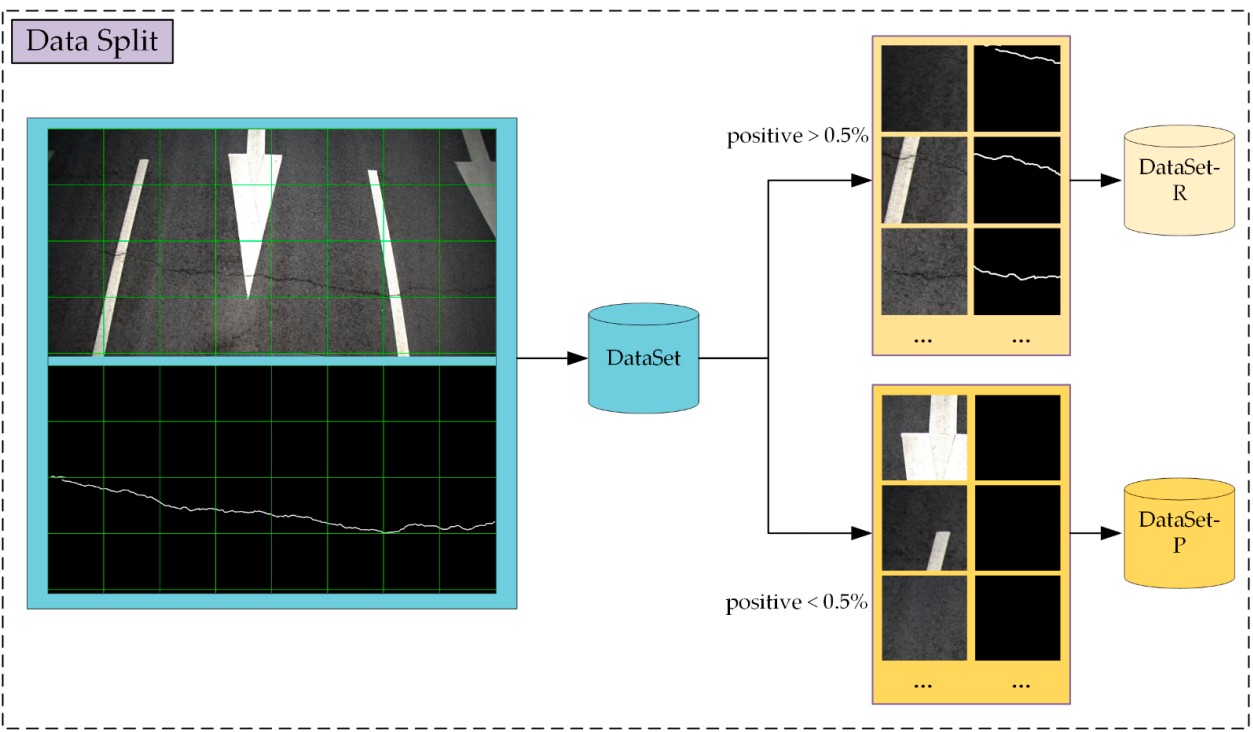

**Figure 2.** The dataset is divided into dataset R and dataset P.

### 3.4. Precision–Recall Flow Loss Method

At the feature level, researchers usually solve the imbalance problem by adjusting the structure or weight of the employed loss function. The Dice loss [31] directly maximizes the F1 score as the goal, which can ameliorate the imbalance between precision and recall but easily leads to gradient disappearance or explosion, especially when the given data are extremely unbalanced. The Focal loss [30] improves upon the BCE loss in terms of stability and robustness, but the hyperparameters need to be set according to expert experience. The hyperparameters are different for diverse networks and datasets. They cannot be universal in various scenes and cannot be adaptively modified during the training process.

To overcome the above problems, we propose an adaptive loss function named the drop loss, which is an improvement of the BCE loss. The definition of the BCE loss is:

$$L_i = [y_i \cdot \log(P_i) + (1 - y_i) \cdot \log(1 - P_i)] \tag{5}$$

where i is the pixel index, $P_i$ is the probability that pixel i is predicted as belonging to the positive category, $y_i$ is the true probability that pixel i belongs to the positive category, and $L_i$ is the BCE loss value generated on pixel i:

$$BCELoss = -\frac{1}{N}\sum_{i=1}^{N} L_i \tag{6}$$

where *N* is the total number of pixels in the batch.

A balanced loss weight and intensity coefficient $\beta$ is introduced:

$$\beta = |PRF| \quad \beta \in [0,1] \tag{7}$$

The gradient caused by positive features is dropped when the recall is stronger, and the positive features are dropped in the opposite case:

$$C_i = \begin{cases} y_i \cdot \log(P_i) \cdot \beta & PRF < 0 \\ (1 - y_i) \cdot \log(1 - P_i) \cdot \beta & PRF > 0 \end{cases} \tag{8}$$

where $C_i$ is the loss value after features dropped on pixel i.

It is desired that the loss function has some randomness to enhance the ability of the neural network to move away from local optimal solutions. A random tensor r is introduced to improve diversity. The tensor $r$ has the same shape as the tensor label of the input neural network and is populated by random numbers that are uniformly distributed on the interval [0,1]. Each $D_i$ in the drop loss is calculated as:

$$D_i = L_i + C_i \cdot r_i \tag{9}$$

where $D_i$ is the drop loss value after the gradient is dropped on pixel i, $L_i$ is the BCE loss value generated on pixel i, and $r_i$ is a random tensor between 0 and 1.

Finally, the expression for the drop loss is:

$$DropLoss = \begin{cases} -\frac{1}{N} \sum_{i=1}^{N} [y_i \cdot \log(P_i) \cdot (1 + \beta \cdot r_i) + (1 - y_i) \cdot \log(1 - P_i)] & PRF < 0 \\ -\frac{1}{N} \sum_{i=1}^{N} [y_i \cdot \log(P_i) + (1 - y_i) \cdot \log(1 - P_i)] & PRF = 0 \\ -\frac{1}{N} \sum_{i=1}^{N} [y_i \cdot \log(P_i) + (1 - y_i) \cdot \log(1 - P_i) \cdot (1 + \beta \cdot r_i)] & PRF > 0 \end{cases} \tag{10}$$

where i is the pixel index, $P_i$ is the probability that pixel i is predicted as belonging to the positive category, $y_i$ is the true probability that pixel i belongs to the positive category, $N$ is the total number of pixels in the batch, $\beta$ is the intensity coefficient in Equation (7), and $r_i$ is a random tensor defined in Equation (9).

By discarding part of the gradient, the precision and recall are in dynamic equilibrium in unbalanced situations, as shown in Figure 3. The drop loss can balance the loss weight at all stages of the training process automatically.

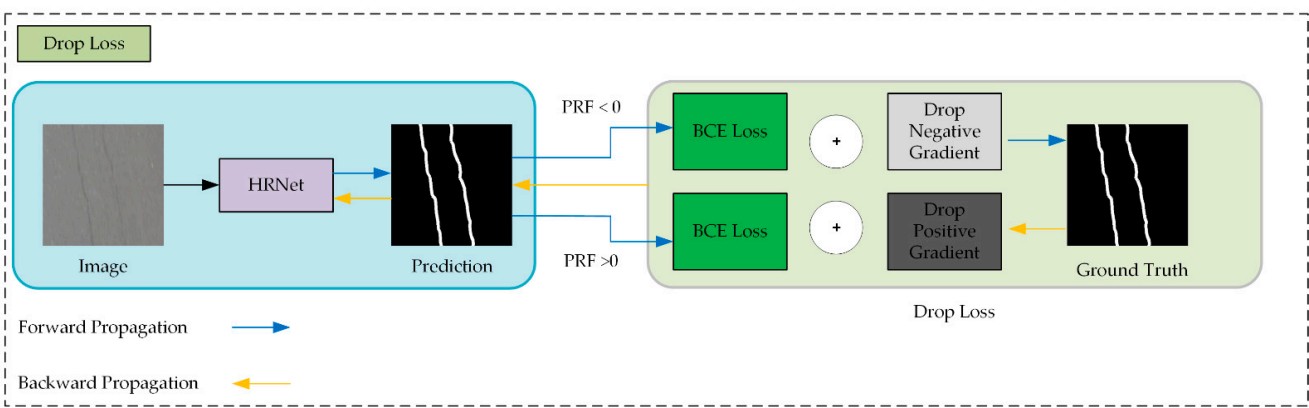

**Figure 3.** Schematic diagram of the drop loss.

## 4. Experimental Details

### 4.1. Dataset Preparation

Although a number of open-source road crack segmentation datasets already exist in the academic community, these datasets have certain limitations.

- Insufficient quantity: To date, most datasets used by scholars contain only a few hundred images, which is far from sufficient for deep learning. For example, the MS-COCO-2018 dataset [39] contains 118,287 images for training, and the ILSVRC-14 dataset contains 456,567 for training. By comparison, the datasets for road crack segmentation are two orders of magnitude smaller. The use of a small amount of data easily leads to the occurrence of overfitting, even early in the training process, and it is difficult to verify the capability of a network on a small dataset. Therefore, a new dataset with a large amount of data is urgently needed.
- Discontinuous sampling: Most existing datasets only include images of very small areas where the roads are damaged and ignore the large areas surrounding healthy road surfaces. When only a small number of normal road surfaces are involved in the training process, the model will misidentify normal objects such as road markings, road facilities, and automobile parts as road cracks.
- Lack of resolution: Most existing datasets are collected by mobile phone cameras or ordinary civilian cameras, whose photosensitive elements are unable to obtain sufficiently clear photos while driving on highways.
- Single shooting time: A dataset obtained entirely at one time cannot provide sufficient diversity for complex external elements such as external light, road shadows, road stains, and road materials.

In view of the above defects, existing datasets cannot be used to verify the performance of neural networks in real road damage inspection scenarios.

We propose a new dataset: the high-resolution road crack dataset (HRRC). The resolution of each HRRC image is 12 million pixels with a width of 4096, a height of 2080, and 5 fps. The dataset contains multiple disturbances, such as overexposure, road markings, road materials, road stains, and shadows. Experts have carefully labeled each image, creating semantically segmented labels. Each large image is divided into 512 × 512 slices, and a total of 24,704 images are obtained. Figure 4 shows examples of samples belonging to the HRRC dataset.

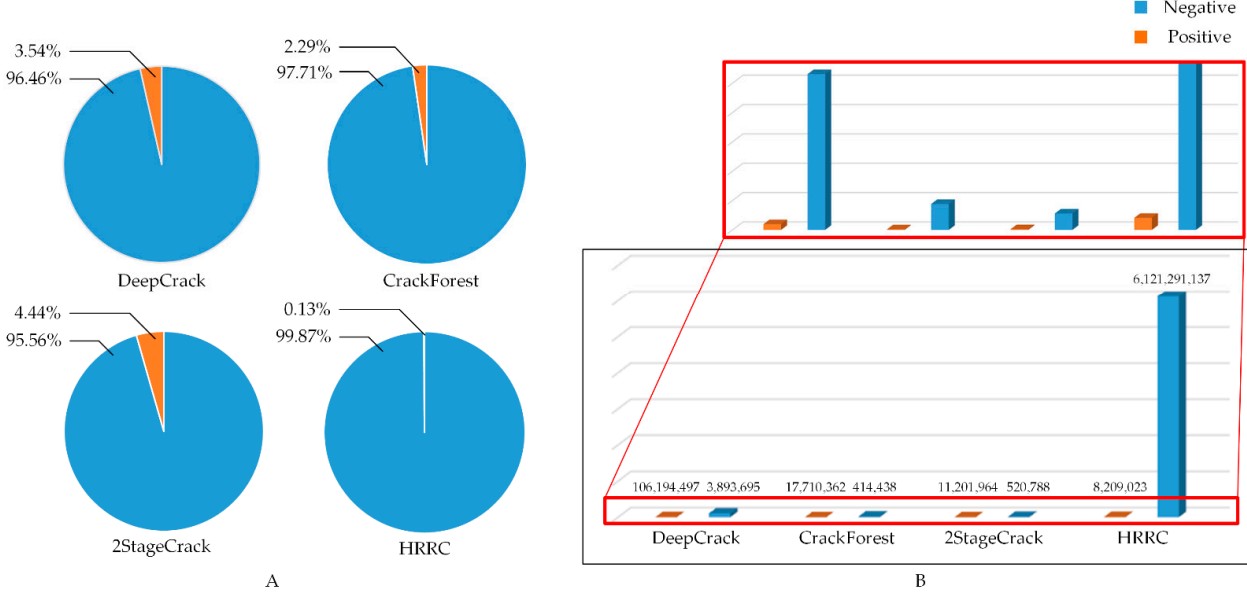

**Figure 4.** The total amount of data in each dataset, as well as their positive and negative sample ratios. (**A**) Positive and negative sample ratios. (**B**) Number of pixels in each dataset.

As shown in Figure 4, the HRRC dataset is the largest road crack dataset at present. A high-resolution Complementary Metal-Oxide-Semiconductor (CMOS) sensor ensures the integrity of various road elements, such as lines and vehicles. Millimeter-level cracks can be seen clearly, and multiple sections of expressway with different materials can be photographed under different weather and lighting conditions. Therefore, this dataset is a high-quality dataset that alleviates the above problems to a certain extent. However, a higher complexity and a smaller positive sample ratio make it difficult for a model to converge and impose higher requirements on the robustness of the model and the advancement of the training method.

HRRC dataset images were captured continuously during the automobile traveling process by an industrial camera, as shown in Figure 5.

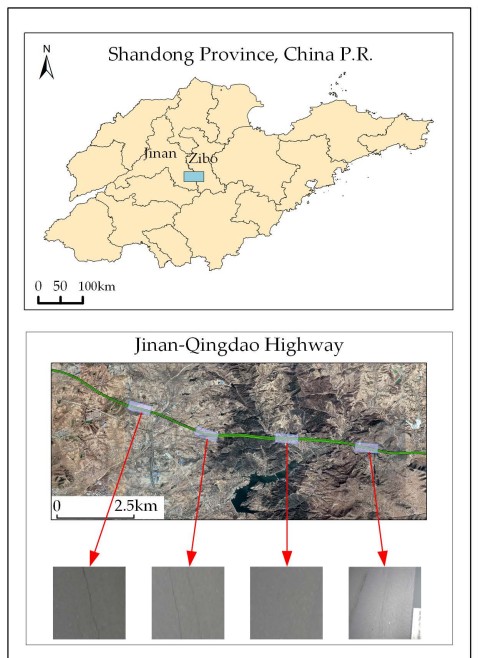 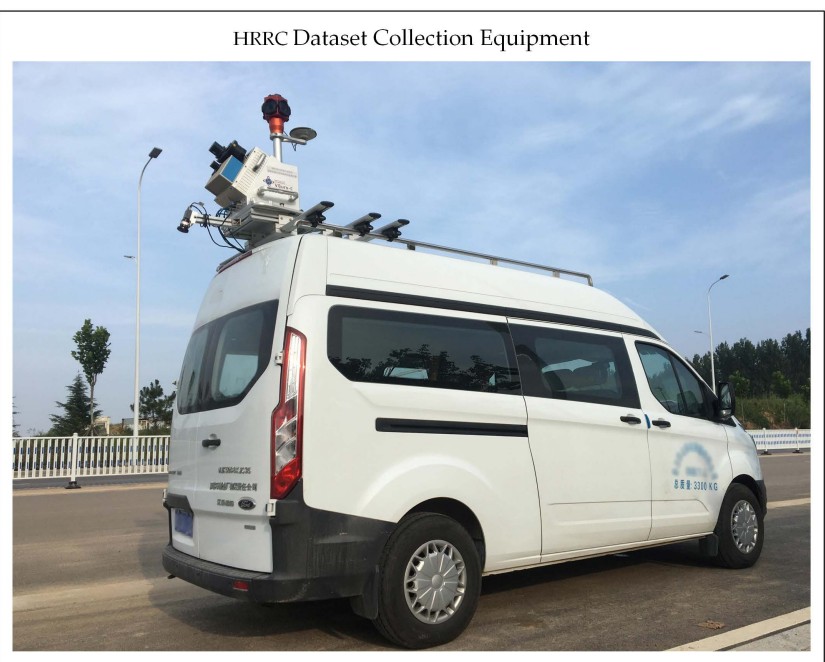

**Figure 5.** Geographical location distribution and collection equipment of the HRRC dataset.

### 4.2. Implementation Environment

All the experiments were based on the following hardware environment: an i7 9700 CPU, an Nvidia GeForce GTX2080ti GPU, and 64 GB of RAM. The software environment is based on Python 3.8, PyTorch 1.6, OpenCV 4.5, and the Windows 10 operating system. The training parameters of the model were set as follows: the learning rate is 0.001, the training batch size is 8, and the validation batch size is 1. All experiments were conducted based on the HRRC dataset only.

### 4.3. Implementation Details

For the first set of experiments, we aimed to find the most suitable convolutional neural network architecture for the road crack segmentation task. Before training, the entire dataset was divided into training and validation sets at a ratio of 9:1. In addition, we used 100 unsliced image-labeled data as an independent test set. The training set was randomly shuffled before breaking it up in order to ensure that each image has the same probability of being input into the network. Due to the large number of images in the dataset, data augmentation was not used to extend the amount of data. The BCE loss function and Adam optimizer were used during the training process. The experiments were conducted with the DeepLabV3+, UNet, UNet++, PSPNet, DeepCrack, FPN, and HRNet frameworks, with 100 training epochs for each network.

To verify the precision of the network architecture, MobileNet_v2 was used as the backbone under the DeepLabV3+, UNet, UNet++, PSPNet, and FPN architectures, and HRNet was also trained using HRNet-18w-small with minimal parameters. The hyperparameters of the model remained the same as in the original paper except for the backbone. The reasons for this are as follows:

(1) MobileNet_v2 has good performance and can be seamlessly integrated with the above network architectures. MMSegmentation [40], PaddleSeg [41], segmentation_models [42], and other standardized open-source code suites support this network, making it convenient for researchers to reproduce.

(2) A network with MobileNet_v2 easily converges and has a fast training speed. This significantly reduces the time required for training when faced with large dataset training tasks.

(3) Regarding the application requirements of a lightweight network, to ensure full coverage of the road surface during shooting, the shooting frequency of the vehicle camera is five shots per second, which leads to the actual application of the data volume being massive. A lightweight network can ensure that staff obtain real-time inspection results.

In addition, we note that batch normalization, which is widely used in computer tasks, facilitates network training. However, the mean and variance obtained are not representative of the global dataset because the batch size used during the training process is not sufficiently large. The variances between different kinds of samples in a small region are large, and the use of a batch size of one during validation and testing causes the batch normalization layer to be more biased to images in which the large class is located; it cannot be well normalized to the regions where the small class is located, thus reducing the recognition ability of the model for the small class. Therefore, this mechanism reduces the recognition ability of the network in the face of unbalanced datasets, so the batch normalization process in HRNet is replaced with group normalization (HRNet-GN), which is the most suitable technique for the road crack segmentation task, to mitigate this problem. The structure of HRNet is shown in Figure 6.

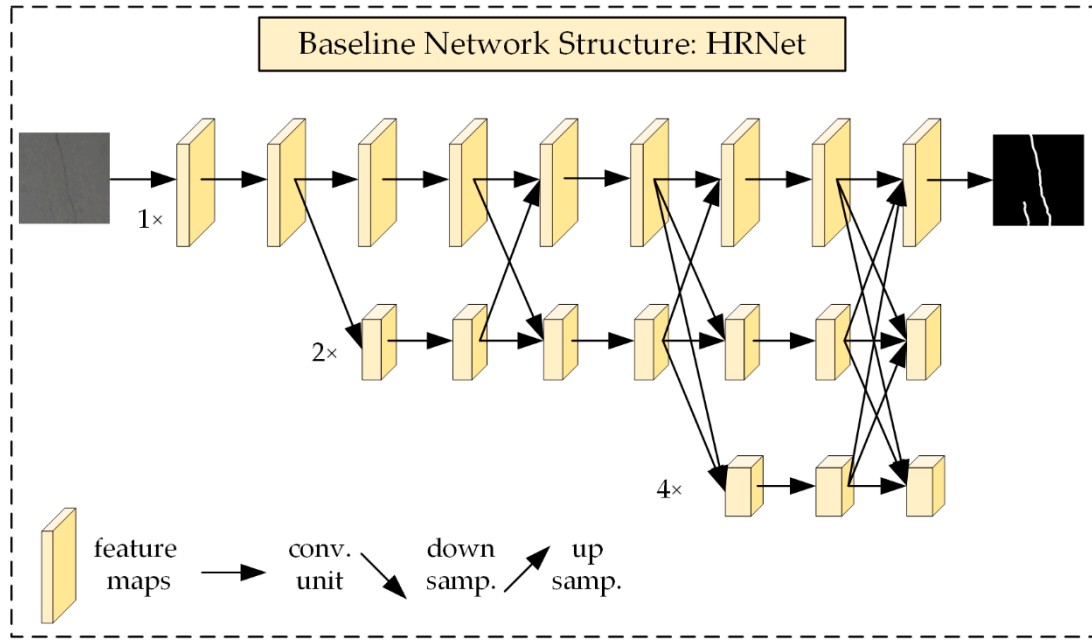

**Figure 6.** HRNet structure schematic ($1\times$, $2\times$, $4\times$ are sampling scales).

The goal of the second set of experiments was to verify the ability of each loss function to deal with the imbalance problem at the feature level. This set of experiments used HRNet-

GN, which is the best network in the first set of experiments, as the baseline. Comparisons were made between the BCE loss, Dice loss, Focal loss, and drop loss. The effectiveness of the drop loss was verified with the same data, network, and training parameters in the road crack segmentation task.

The goal of the third set of experiments was to verify the generality and advancement of the PRF method proposed in this paper, which includes direction determination, intensity evaluation, sampling, and loss weight adjustment. This method adaptively adjusts itself throughout the training process and can be integrated into all convolutional neural networks without adding extra parameters. The experiments were first conducted to evaluate the performance improvement yielded by this method for convolutional neural networks by conducting ablation tests on the sample and loss methods with HRNet-GN as the baseline. Second, alignment tests were also conducted on HRNet to verify the generality and universality of this method.

Under the PRF sampling method, the dataset was divided into two sub-datasets. The probabilities of images from different datasets were changed, and the sampling method was also changed from non-release sampling to release sampling. The number of images in dataset R was used as the dataset length in these experiments, resulting in a smaller number of samples per epoch. To fully train the model, we adjusted the number of epochs to 800.

### 4.4. Methods for Evaluation

To objectively evaluate the performance of the network, the precision, recall, and IOU metrics were used to verify the performance of the semantic segmentation model. Their definitions are as follows:

$$precision = \frac{TP}{TP + FP} \tag{11}$$

$$recall = \frac{TP}{TP + FN} \tag{12}$$

$$IOU = \frac{TP}{TP + FP + FN} \tag{13}$$

$$F1score = \frac{2TP}{2TP + FP + FN} \tag{14}$$

$TP$, $FP$, and $FN$ are the numbers of true-positive, false-positive, and false-negative pixels, respectively. The threshold for splitting positive and negative in a predicted probability map is 0.5.

## 5. Results

In this section, we present and analyze the experimental results in three parts according to the experimental design.

### 5.1. Network Structure Experiment Results

The first set of experiments involving different network architectures is shown in Table 1. Surprisingly, the DeepLabV3+ results are the worst, and this result does not match the performance of DeepLabV3+ in natural image semantic segmentation. To explain this phenomenon, we believe that dilated convolution expands the view of the convolution but reduces the spatial resolution of the feature layers. This approach can be advantageous in dealing with large targets but cannot cope well with road cracks that are only a few pixels in width. Apart from DeepLabV3+, the precision rankings of the other networks do not differ much from their performance in natural images. HRNet performs the best and leads the other network architectures by a large margin. It is worth mentioning that the IOU of HRNet-GN improves by 4.6%, indicating that group normalization has a significant advantage over the batch normalization structure when facing imbalance problems. The

experimental findings indicate that the HRNet structure had outstanding performance and that group normalization had an advantage over batch normalization in this task.

**Table 1.** Scores of different network structures trained with the BCE loss.

| Network Structure | IOU (%) | F1 (%) | Recall (%) | Precision (%) |
|---|---|---|---|---|
| DeepLabV3+ [26] | 21.79 | 30.45 | 24.27 | 86.6 |
| U-Net [17] | 23.93 | 33.79 | 25.46 | 89.35 |
| PSPNet [22] | 25.19 | 34.26 | 34.14 | 78.85 |
| UNet++ [20] | 27.69 | 36.74 | 31.08 | 88.75 |
| DeepCrack [9] | 34.7 | 48.69 | 60.97 | 46.32 |
| FPN [23] | 35.61 | 46.53 | 41.6 | 84.42 |
| HRNet [11] | 40.65 | 52.21 | 49.99 | 79.57 |
| HRNet-GN | 44.61 | 56.53 | 58.65 | 75.37 |

Table 2 shows the parameters, multiply-accumulate operations (MAdd), floating point operations (Flops), and GPU memory values of different network structures for inference purposes. Among the tested methods, PSPNet has the smallest number of parameters, computational complexity, and GPU memory needed, but its precision is 22.27-percentage points lower than the best precision achieved by HRNet-GN. The advantage of small computation complexity does not offset the excessive capacity degradation of PSPNet. The DeepCrack structure has the largest number of parameters and MAdd value, but its IOU is 7.53-percentage points lower than that of HRNet-GN, so it is not a good choice. The number of HRNet parameters is only larger than that of PSPNet, but its MAdd, Flops, and GPU memory are the second largest—only being smaller than those of DeepCrack. This is because the HRNet structure retains more feature maps with different resolutions and requires more data operations and memory allocations within the network to achieve cross-resolution fusion with the highest parameter utilization rate. Considering the precision, number of parameters, Flops, and memory requirements, we believe that HRNet-GN has the highest precision and fewest parameters and is the most suitable convolutional neural network structure for road crack semantic segmentation applications at this stage.

**Table 2.** Information about number of parameters, number of calculations, and GPU memory usage levels of the tested networks.

| Network Structure | Parameters | MAdd | Flops | GPU Memory |
|---|---|---|---|---|
| DeepLabV3+ [26] | 4,378,513 | 2.3 G | 1.16 G | 112.66 MB |
| U-Net [17] | 6,628,945 | 5.16 G | 2.59 G | 124.13 MB |
| PSPNet [22] | 94,969 | 305.55 M | 157.82 M | 59.88 MB |
| UNet++ [20] | 6,824,721 | 6.85 G | 3.43 G | 157.67 MB |
| DeepCrack [9] | 14,720,389 | 30.76 G | 15.39 G | 155.87 MB |
| FPN [23] | 4,215,425 | 3.7 G | 1.86 G | 88.92 MB |
| HRNet [11] | 3,934,629 | 10.62 G | 5.34 G | 222.56 MB |
| HRNet-GN | 3,934,629 | 10.62 G | 5.34 G | 222.56 MB |

*5.2. Experimental Loss Function Comparison Results in a Case with Imbalance*

Table 3 shows the comparison between the BCE loss, the Dice loss, the Focal loss, and our drop loss. The Dice loss is unable to learn enough positive features, resulting in a failure to converge. Without parameter tuning, the Focal loss with its default parameters ($\gamma = 2$, $\alpha = 0.25$) is even worse than the BCE loss for the road crack segmentation task. From the experimental results, it can be concluded that the poor performance of the Dice and Focal losses is caused by the fact that the recall is greatly below the precision in the case of extreme imbalance among the HRRC data categories. The BCE loss achieves good results and stable convergence with no need to adjust parameters, as well as low training difficulty. Supported by the recurrent adaptive network framework, the drop loss inherits these advantages and yields an exciting achievement: a 9.35% IOU increase.

**Table 3.** Comparison among the Dice loss, Focal loss, and drop loss.

| Network Structure | IOU (%) | F1 (%) | Recall (%) | Precision (%) |
|---|---|---|---|---|
| DICE loss | $2.73 \times 10^{-9}$ | $2.73 \times 10^{-9}$ | $2.73 \times 10^{-9}$ | 100 |
| Focal loss | 25.19 | 48.59 | 34.14 | 78.85 |
| BCE loss | 44.61 | 56.53 | 58.65 | 75.37 |
| Drop loss | 53.39 | 66.46 | 71.16 | 71.72 |

*5.3. Ablation Experiments Involving the PRF Method*

Ablation experiments are also performed on both HRNet and HRNet-GN to verify the reliability of the proposed method. According to the test results in Table 4, the following conclusions can be drawn:

- Both PRF sampling and the drop loss method can clearly increase the IOU score: 13.1% on HRNet and 9.56% on HRNet-GN.
- The recurrent adaptive network framework minimizes the performance gap between HRNet and HRNET-GN from 3.96% to 0.41%.
- The drop loss method applied to the feature yields slightly higher results than those of the sampling method applied to the data layer.

**Table 4.** Ablation experiment results obtained based on HRNet and HRNet-GN.

| Network Structure | IOU (%) | F1 (%) | Recall (%) | Precision (%) |
|---|---|---|---|---|
| HRNet | 40.65 | 52.21 | 49.99 | 79.57 |
| HRNet + PRF sampling | 52.78 | 66.19 | 70.98 | 68.01 |
| HRNet + Drop loss | 53.25 | 66.75 | 70.11 | 70.37 |
| HRNet + PRF sampling + Drop loss | 53.63 | 67.28 | 71.22 | 69.16 |
| HRNet-GN | 44.61 | 56.53 | 58.65 | 75.37 |
| HRNet-GN + PRF sampling | 53.33 | 66.61 | 70.01 | 71.56 |
| HRNet-GN + Drop loss | 53.39 | 66.46 | 71.16 | 71.72 |
| HRNet-GN + PRF sampling + Drop loss | 54.03 | 67.39 | 69.77 | 71.96 |

With the recurrent adaptive network framework, HRNet-GN obtains an IOU score of 0.5416, which is the state-of-the-art value on the HRRC dataset (The codes used in this work can be accessed via the URL provided in the Supplementary Materials section).

Figure 7 shows the variations in precision and recall with the IOU and PRF values during the training processes of the HRNet-GN and HRNet-GN+ the recurrent adaptive network framework. By observation, we find the following:

- Convergence acceleration: In Figure 7f,g, the training process starts with a precision greater than 0.98 and a recall less than 0.02, and this phenomenon lasts for five epochs before it starts to improve. In Figure 7b,c, the model starts to converge well at the second epoch, and the IOU value is greater than 0.5 in the third epoch in Figure 7a, while in Figure 7e reaches the same level after 50 epochs.
- Balancing precision and recall: In Figure 7g, the maximum value of recall is less than 0.6, and the minimum value of precision is greater than 0.75 throughout the training process in Figure 7f. This large gap continues over the entire procedure. In Figure 7d, the recurrent adaptive network framework starts to work at the second epoch, and the gap between precision and recall decreases rapidly so that it remains stable at approximately 0.1; this is an obvious reduction in the precision–recall gap.
- Adaptive adjustment throughout the whole process: At the beginning of training, the first epoch is used for pre-training to obtain the first PRF value, and from the second epoch until the end of training, the PRF value is adaptively updated after a specific interval (five epochs in these experiments). This mechanism makes the sampling and loss weights more suitable for the next training stage and adjusts the training parameters according to the model states observed throughout the process.

- Significant performance improvement: As training proceeds, the IOU and PRF curve amplitudes gradually decrease. This means that the PRF framework makes the training process stable and reduces the training difficulty. More importantly, with the optimized training of the recurrent adaptive network, the IOU value of the model greatly improves, which significantly improves the performance of the model in the face of unbalanced datasets. Figure 8 shows that PRF sample and drop loss method in recurrent adaptive network are used together to obtain the best performance.

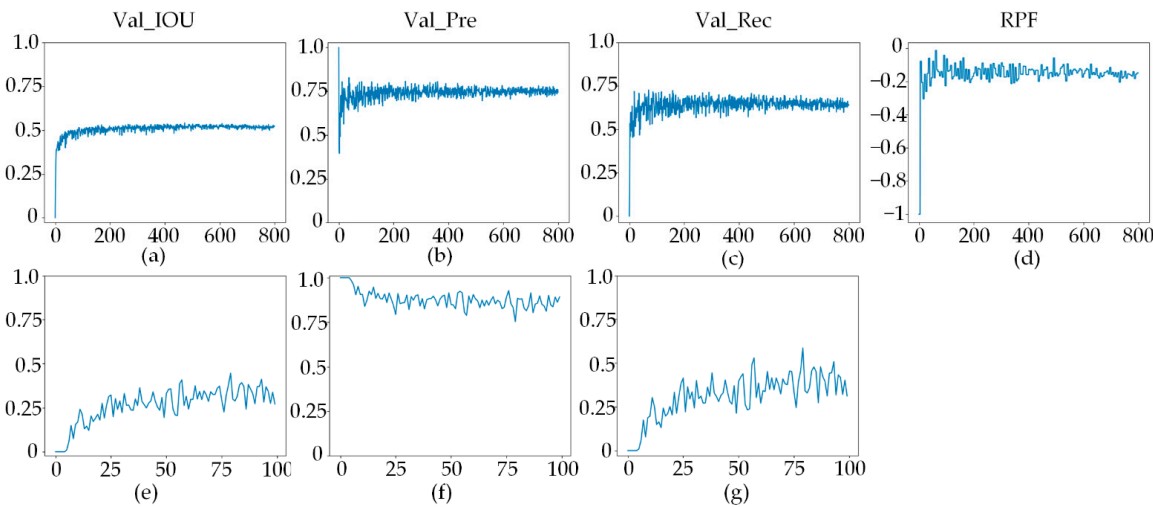

**Figure 7.** Variations in the precision, recall, and IOU curves during training. (**a**–**c**) show validation IOU, precision and recall when training whit recurrent adaptive network. (**d**) shows values and changes of PRF during training. (**e**–**g**) show validation IOU, precision and recall with normal training.

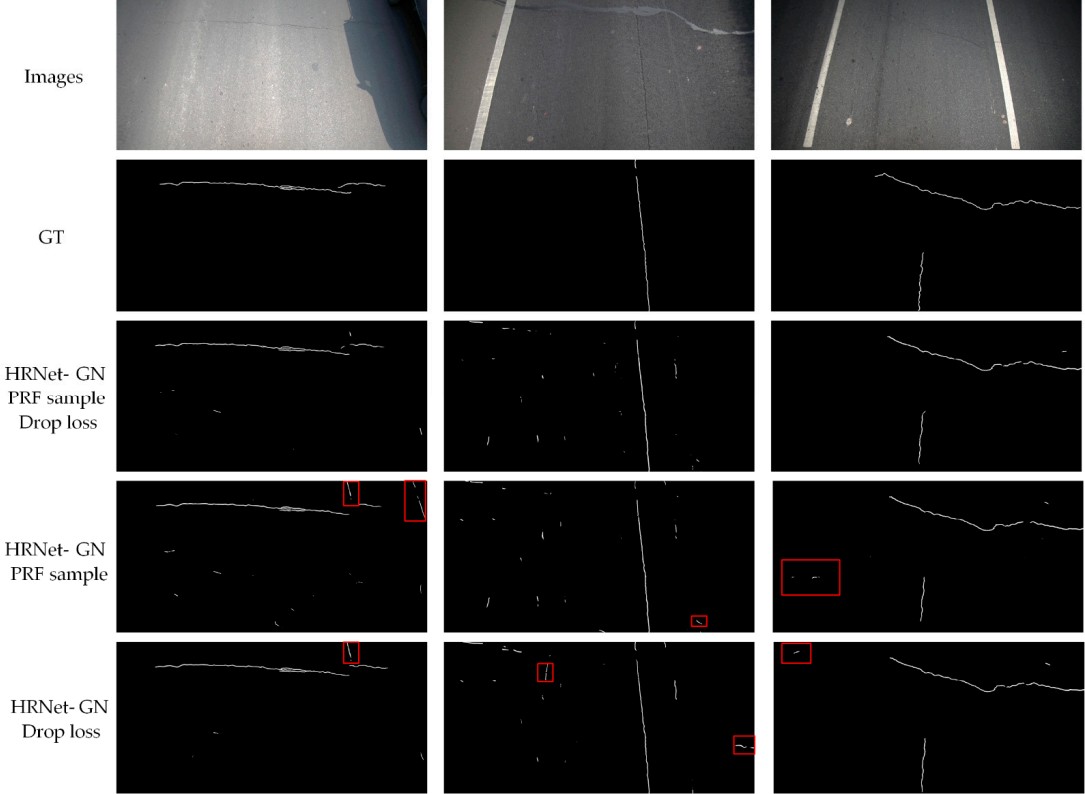

**Figure 8.** Results of ablation experiments (red boxes represent the regions of poor performance that model obtained from the single method training in the ablation experiment).

## 6. Discussion

It is well known that the performance of a model can be improved by adjusting the sampling and loss weights in a suitable way when faced with data imbalance. The most interesting finding was that we guided the direction and value of parameter tuning in sampling and loss by measuring the deviation between precision and recall, and we repeated the process at regular intervals during training. We refer to this training framework as a "recurrent adaptive network".

In this study, there are two key points: (1) The large gap between precision and recall in the road crack segmentation task is an external manifestation of positive and negative sample imbalance. Their changes need to be analyzed to guide the adaptive adjustment of the balance parameters. (2) The PRF keeps changing during the training process, which indicates that the negative impact of positive and negative sample imbalance varies at different stages of the training process. The parameters need to be adjusted periodically as the training progresses.

In previous studies, no researcher has demonstrated that narrowing the deviation between precision and recall can improve model performance. What is surprising is that the experimental results in this paper demonstrate that this approach is effective in solving the imbalance problem in the road crack segmentation task. The training converges faster and leads to an improved model performance, which is supported by the construction of a recurrent adaptive network.

Of course, there are shortcomings and limitations in our work. The most effective way to solve the imbalance problem is to increase the number of samples in small categories, which can be achieved by GAN networks, but this is not considered in this method. In the experiments described in this paper, the precision was higher than the recall throughout the training process, which shows that the imbalance is only mitigated and not eliminated. The desired state is that precision and recall curves interpolate with each other with oscillation, but it is obvious that the experiment does not meet this expectation.

In the next study, we will try to continue our research by using a GAN to generate simulated data based on the framework of this paper and using the device to collect more positive samples to alleviate the degree of imbalance between positive and negative samples. Moreover, the question of how a complete classification can be provided from the smaller frames for segmentation needs to be addressed in our next research. In addition, there is road crack incompleteness in the prediction results, and this study does not improve the completeness of the small category targets.

## 7. Conclusions

In this study, a new high-resolution road crack database (HRRC) was produced; it came from a real application scene, and data were collected by a vehicle-mounted industrial camera. Currently, HRRC is the largest and most complicated dataset in the field of road crack research. We analyzed semantic segmentation frameworks based on convolutional neural networks, and performance tests were conducted on HRRC. The results show that HRNet-GN is the most suitable convolutional neural network framework for road crack segmentation. A recurrent adaptive network framework based on the idea of precision–recall flow, including flow direction determination, intensity estimation, sampling, and drop loss calculation, is proposed. This framework measures the deviation between precision and recall to determine the degree of observed imbalance. Regarding the data-level and feature-level aspects, this imbalance is automatically bridged to obtain a better trained model. This framework maintains a balance between precision and recall at the data and feature levels and performs well in training tasks with extremely unbalanced datasets, providing adaptive tuning, accelerated network convergence, balanced precision and recall, and significant performance gains. As a result, the recurrent adaptive network achieves a state-of-the-art performance on this dataset, with an F1-score of 56.53, which is far better than the other methods.

**Supplementary Materials:** The experimental codes can be found here: https://aistudio.baidu.com/aistudio/projectdetail/4242896?contributionType=1&shared=1 (accessed on 19 June 2022).

**Author Contributions:** Conceptualization, Y.Z. and J.F.; methodology, Y.Z. and J.F.; software, Y.Z.; validation, M.Z., R.L. and Z.S.; formal analysis, Y.Z.; investigation, R.L.; resources, M.Z. and Z.S.; data curation, Y.Z. and R.L.; writing—original draft preparation, Y.Z. and J.F.; writing—review and editing, J.F. and B.G.; visualization, M.Z. and Z.S.; supervision, J.F.; project administration, J.F.; funding acquisition, J.F. All authors have read and agreed to the published version of the manuscript.

**Funding:** This research was funded by the National Natural Science Foundation of China (grant nos. 42171413, 42101306 and 42001414), a grant from the State Key Laboratory of Resources and Environmental Information System, the Shandong Provincial Natural Science Foundation (grant nos. ZR2020MD015, ZR2020MD018 and ZR2019BD033), the National Key Research and Development Program of China (grant no. 2017YFB0503500), and the Young Teacher Development Support Program of Shandong University of Technology (grant no. 4072-115016).

**Conflicts of Interest:** The authors declare no conflict of interest.

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
