# Peer review of "A Recurrent Adaptive Network: Balanced Learning for Road Crack Segmentation with High-Resolution Images"

_remotesensing, doi:10.3390/rs14143275_

Round 1

Reviewer 1 Report

Dear Authors,

I had a real pleasure reading your manuscript. You did interesting research and revealed substantial knowledge both in the fields of image analysis and the application aspect of the matter. I have to admit, I am not fully convinced of the correctness of the reference balancing recall and precision to the heat flow process. Regardless, the proposed idea is undoubtedly interesting, and the article well illustrates its effectiveness. Therefore I strongly opt for publishing it in the Remote Sensing journal. However, before it is possible, a few minor corrections have to be done. I put my remarks in the list below.
1. In line 23 (abstract): Do you really mean accuracy and recall? You balance the precision-recall in the rest of the paper.
2. line 40: canny should start with a capital letter
3. The positive and negative cases (paragraph in lines 64 - 68) should be more clearly described: do you mean pixels or whole images? It is not clear to the reader.
4. You lack an explanation of abbreviations used in the manuscript, it has to be corrected before publication.
5. You lack the explanations of the symbols in the equations. It has to be corrected before publication (there should not be any symbol without explanation and there are many like that in the current version of the manuscript).
6. More explanation on tensor r (line 312 and equation 9) is needed. What does it mean that it is random? What probability distribution was used to generate its value, and why? Please, be more specific.
7. Reference no. 10 is broken, there is no title of the cited paper.

Reviewer 2 Report

This paper proposes a novel road crack dataset and compares different semantic segmentation models. I believe the authors brought too much attention to a new sampling technique that is quite confusing and not well funded with the literature. There are not enough comparisons with other simple and effective methods, such as assigning class weights in the loss functions. There is a misplacement of information within the sections and a lack of crucial topics, such as discussion. My major concerns are as follows:

1 - The most consistent way of analyzing supervised deep learning-based models is to separate the dataset into training, validation, and testing. Using this configuration allows to save the best model of the validation set (using whatever criterion the authors want) and use it in an independent test set. Besides, it brings more reliability to the results.

2 – The placement of certain information does not seem coherent in some scenarios. For example, the network architectures that were used would be better suited in the materials and methods section. On the other hand, the description of different datasets used for road cracks, and their studies, could be placed in the related works section.

3 – This paper presents no discussion. It is very important to state comparisons and insights in this section.

4 – (Line 95-96) It is not appropriate to cite the results of this paper in the introduction.

5 – This work would bring much more reliability if the source code and dataset were made publicly available.

6 – This dataset brings a binary classification problem. The semantic segmentation outputs a probability value to each pixel, and to extract the precision or recall; it is necessary to define a threshold. This work does not even cite the threshold used. Another problem is that adjusting a higher precision and recall can be done at a certain point by adjusting this threshold value.

7 – the citations should present a space. For example, change “DeepLabV3+[25]” to “DeepLabV3+ [25]”. Adjust this problem for all citations.

8 – the comparison with the second law of thermodynamics only brings confusion to the paper. The entire description of the “flow” mechanism is unclear, not funded by previous research, and the way it is explained is not convincing. Several other sampling strategies can bring balanced data. For example, even if considering “positive” frames in the created dataset, the imbalance at a pixel level would still be big.

Some notes on this topic

(Lines 252-253) “if positive samples dominate, the recall will be greater than the precision. If negative samples dominate, the situation will be the opposite”. This affirmation is not correct.

(Lines 260-262) It is not appropriate to cite results in this section. This section should focus only on the methods, not on any sort of results. This brings confusion to the readers.

The authors mention the usage of weights, for example, in the binary cross-entropy loss. It would be interesting to see results using the weights. Adjusting the class weights, in many cases, is sufficient. There are many ways to choose weights, which do not to be manual. The authors can automatically calculate the proportions of positive and negative pixels before training the model.

10 – A crucial part of the proposed solution is explaining how you would provide a complete classification from the smaller frames. Many remote sensing studies face this problem, for instance, semantic and panoptic segmentation. This would be a good topic for discussion in this paper.

11 – The number of references in this paper should increase.

12 – The model comparison comprises parameters that were not explained. For example, the DeepLabv3+ has atrous rates that can be modified to increase the metrics significantly.

13 – (Line 386) “data augmentation is not used to extend the amount of data”. Why not use random flips? This strategy would not increase the number of samples.

Reviewer 3 Report

The paper investigates a RAN to balance the learning scheme for crack detection. Overall, the pape is well written. However some points need to be addressed to further enhance the quality ofof the manuscript.

1 - please mention your main finding from the results (figures/numbers) in the abstract.

2- please mention the contribution and novelty clearly at the end of the introduction.

3- for imbalance data, there are recent works done trying to improve the capability of machine learning models. You can refer to them in your literature to further

improve the literature. Such techniques were also able to improve the accuracy.

https://doi.org/10.3390/rs13194011

https://doi.org/10.1049/itr2.12146

doi:10.3390/infrastructures5070061

4- In the discussion, you can mention the limitations of your method and also c

Compare it with min3 recent  works that have similar scope.

And then you can mention how your research is better than theirs. You can add a table for this purpose of comparison.

5- please add the only important results numbers

in the conclusion. You can add recommendations for future works.

Round 2

Reviewer 2 Report

The authors have made the changes.

Author Response

Please see the attachment. Thank you again for your valuable comments!

Reviewer 3 Report

1- The manuscript has been improved. You do not need to delete the organization of your paper from the introduction (You can keep it).

"The remainder of this paper is organized as follows. In section II, we introduce 121 some existing convolutional neural network structures for segmentation and methods 122 for solving the imbalance problem. Section III describes the proposed method. The ex- 123 perimental results are reported in section IV, and this paper is concluded in section V. "

2- English proofreading is required.

Author Response

(The authors gave the same response as above.)
